# Role of Ultrasound Methods for the Assessment of NAFLD

**DOI:** 10.3390/jcm11154581

**Published:** 2022-08-05

**Authors:** Golo Petzold

**Affiliations:** Department of Gastroenterology and Gastrointestinal Oncology, University Medical Center Goettingen, 37075 Goettingen, Germany; golo.petzold@med.uni-goettingen.de

**Keywords:** nonalcoholic fatty liver disease (NAFLD), fibrosis, steatosis, hepatocellular carcinoma, ultrasound, elastography, 2-dimensional shear wave elastography (2DSWE), transient elastography

## Abstract

Nonalcoholic fatty liver disease (NAFLD) is the most common liver disease worldwide. The prevalence in patients with type 2 diabetes mellitus is between 55–80%. The spectrum of NALFD ranges from simple steatosis to aggressive steatohepatitis with potentially progressive liver fibrosis up to cirrhosis and hepatocellular carcinoma. In clinical practice, there are two important aims: First to make the diagnosis of NAFLD, and second, to identify patients with advanced fibrosis, because extent of fibrosis is strongly associated with overall mortality, cardiovascular disease, hepatocellular carcinoma, and extrahepatic malignancy. Histology by liver biopsy can deliver this information, but it is an invasive procedure with rare, but potentially severe, complications. Therefore, non-invasive techniques were developed to stage fibrosis. Ultrasound is the primary imaging modality in the assessment of patients with confirmed or suspected NAFLD. This narrative review focus on different ultrasound methods to detect and graduate hepatic steatosis and to determine grade of fibrosis using elastography-methods, such as transient elastography and 2-dimensional shear wave elastography in patients with NAFLD. Particular attention is paid to the application and limitations in overweight patients in clinical practice. Finally, the role of B-mode ultrasound in NAFLD patients to screen for hepatocellular carcinoma is outlined.

## 1. Introduction

Nonalcoholic fatty liver disease (NAFLD) is the most common liver disease worldwide. The prevalence is between 20–30% and can be considered endemic in developed countries. In patients with type 2 diabetes mellitus, the prevalence is even between 55–80% [1]. Because of this high prevalence, recent guidelines recommend to screen all patients with diabetes mellitus type 2 for NAFLD, especially for advanced fibrosis [2]. NAFLD is defined as abnormal accumulation of fat (>5%) in the hepatocytes in the absence of significant alcohol consumption (>30 g/d in men, >20 g/d in women [3] or defined as ≥14 drinks/week for men or ≥7 drinks/week for women) and exclusion of secondary causes of fatty liver disease (for example, hepatitis C or Wilson disease). Histologically, the amount of fat in the liver is graded as S0, steatosis in <5% of hepatocytes; S1, 5–33%; S2, 34–66%; and S3, >66% [4]. The spectrum of NALFD ranges from simple steatosis to aggressive NASH (nonalcoholic steatohepatitis) with potentially progressive liver fibrosis up to cirrhosis and hepatocellular carcinoma (HCC). 

A recent nationwide study in Sweden revealed that all NAFLD histological stages are associated with significantly increased overall mortality, and this risk increased progressively with increased degree of fibrosis. The excess mortality associated with NAFLD was primarily due to extrahepatic cancer followed by cirrhosis, cardiovascular disease, and HCC [5].

Lifestyle modification is the core of treatment of NAFLD. An effective drug therapy has not yet been approved, but approvals are expected in the near future for numerous ongoing phase II and III studies [3,6]. According to current guidelines, NAFLD patients with cirrhosis and advanced fibrosis (F3) should undergo regular HCC screening. The presence of esophageal varices should also be evaluated in this collective. The main question is how can the stage of NAFLD be determined? The gold standard is liver biopsy. Histology provides information on the extent of steatosis, the degree of inflammation and the stage of fibrosis. In addition, possible differential diagnoses such as autoimmune hepatitis can be ruled out. Liver biopsy is a safe and clinically established method, but as an invasive examination it is associated with potential complications (mortality up to 0.2%). Pain is common (up to 84%), inpatient hospitalization is usually necessary, and patient acceptance is correspondingly low [7,8]. Furthermore, a biopsy cylinder represents only 1 per 50,000 of the entire liver tissue, which is associated with the risk of a “sampling error”. In addition, high interobserver and intraobserver variability has been reported [9,10,11].

In the case of an endemic disease, it is understandable that not every patient with NAFLD or suspected NAFLD can be biopsied and especially not several times during the course of the disease. Biopsy is still essential in NAFLD patients treated in the context of clinical studies. The inclusion criteria and endpoints in these studies are usually defined histologically (steatohepatitis, ballooning, grade of fibrosis).

According to the current EASL (European Association for the Study of the Liver) guideline, magnet resonance imaging (MRI) using proton density fat fraction (PDFF) is now considered equivalent to biopsy to determine the degree of liver fat content [3]. Disadvantages of the MRI are the low availability, the high costs, and the high expenditure of time.

Guidelines recommend using non-invasive blood-based tests to screen patients with NAFLD or suspected NAFLD for advanced liver fibrosis [3]. Best-validated tests are FIB-4 (Fibrosis-4-Index) and NFS (NAFLD Fibrosis Score), which can rule out advanced liver fibrosis with high accuracy. On the other hand, there is a high risk of false positive results for advanced fibrosis and, especially for NFS, a lower performance in obese and diabetic patients is reported [12]. 

The primary imaging modality in suspected NAFLD is sonography. It is inexpensive, widely available, free of side effects, can be repeated as often as required, and has a high level of patient acceptance due to the direct examination by the doctor. Sonography has increasingly developed in recent years, so that in addition to the well-known B-mode sonography, newer techniques, such as elastography or quantitative fat determination, are also available. Our aim is to provide a practice-oriented overview of the use of sonography methods for the assessment of patients with NAFLD.

## 2. Detection and Graduation of Hepatic Steatosis

Transabdominal ultrasound should be used as the primary imaging method in patients with suspected NAFLD [3]. The basic sign for steatosis is the increased echogenicity of the liver parenchyma in comparison to the cortex of the right kidney because intracellular accumulation of fat vacuoles reflects the ultrasound beam. The classification of steatosis is usually graded as follows: grade 0: normal echogenicity of the right liver lobe in comparison with the cortex of the right kidney; grade 1: slight, diffuse increase in fine echoes in liver parenchyma with normal visualization of diaphragm and intrahepatic vessel borders; grade 2: moderate, diffuse increase in fine echoes with slightly impaired visualization of intrahepatic vessels and diaphragm; grade 3: marked increase in fine echoes with poor or non-visualization of the intrahepatic vessel borders, diaphragm, and posterior right lobe of the liver [13]. The sensitivity of B-mode ultrasound to detect hepatic steatosis varies between 53–76%, the specificity is between 76–93%. In presence of sonographic criteria of a higher-grade steatosis (i.e., impaired visualization of diaphragm, Figure 1), the probability of the presence of hepatic steatosis is nearly 100% [14,15,16,17,18]. On the other hand, the sensitivity of B-mode sonography is poor in the case of mildly pronounced steatosis (<20–30%), so that NAFLD cannot be ruled out with certainty if the B-mode criteria mentioned are absent. Detection of focal fatty sparing is associated with an increased attenuation coefficient and is thus an expression of higher-grade hepatic fatty degeneration [19].

Calculation of the hepatorenal index (HRI) based on B-Mode ultrasound images can potentially increase the diagnostic accuracy of detection and graduation of hepatic steatosis. HRI is defined as the ratio of the echo intensities of the liver parenchyma and renal cortex. Diagnostic accuracy of HRI varies widely from study to study, using histology as the reference standard: Sensitivity for detection of steatosis varied between 62.5% and 100%, specificity reached from 54% to 95%. Optimal cut-off values for mild steatosis range from 1.28 to 2.01. The different results of all studies that examined the accuracy of HRI to detect hepatic steatosis indicate that HRI heavily depends on the used ultrasound device and the cause of liver disease. Additionally, in patients with mild steatosis, sensitivity was lower [20,21,22,23]. 

In summary, B-mode ultrasound is an excellent method to detect moderate and severe steatosis in patients with NAFLD. For the diagnosis of mild steatosis, ultrasound and HRI have a low sensitivity, so there is a diagnostic gap in these patients.

To solve this problem, new ultrasound methods were recently developed. The most common method is the ultrasound-based controlled attenuation parameter (CAP), which is integrated into the FibroScan (echosens). The CAP technology has been available since 2010. During liver stiffness assessment with transient elastography, the CAP algorithm calculates the attenuation of the ultrasound signal and is expressed in dB/m, ranging from 100–400 dB/m, so that quantification of hepatic steatosis is possible. CAP shows a good correlation with the biopsy proven hepatic steatosis, and cut-off levels for the four grades of steatosis could be defined. The optimal cut-offs values are 248, 268, and 280 dB/m for identifying steatosis grades > S0, >S1, and >S2, respectively. Nevertheless, sensitivity for discrimination between patients with steatosis and without steatosis is only 68.8%, with a specificity of 82.2% [24]. Similar to B-mode, sonography, especially differentiation between patients without steatosis and those with mild steatosis, is difficult. Furthermore, diabetes and body mass index (BMI) have an independent impact on CAP-value and must be taken into account in the interpretation. In addition, technical limitations of the method are a BMI > 35 kg/m² and a skin-to-liver capsule distance >25 mm. These conditions can lead to an overestimation of the CAP values. To address this problem, XL-probe for obese patients was developed. A large meta-analysis showed a similar diagnostic accuracy of XL-probe for detection of steatosis in obese patients compared with the results of M-probe in non-obese patients. However, the diagnostic accuracy of XL probe seems to be lower in differentiating between no or mild steatosis and higher grades of steatosis [25].

Next to Fibroscan-based CAP-quantification technique, other technologies were developed by different sonography manufacturers to quantitatively assess liver fat content. These techniques are integrated in high-end sonography devices. The methods are based on analysis of the radiofrequency echoes detected by the transducer allowing calculation of parameters for quantifying the fat content in the liver. These new methods include spectral-based techniques and techniques based on envelope statistics. The spectral-based techniques that have been used in clinical studies are those estimating the attenuation coefficient and those estimating the backscatter coefficient [26]. A current position paper of the World Federation for Ultrasound in Medicine and Biology (WFUMB) provides a detailed overview, particularly of the technical aspects [26].

Due to the novelty of the methods, not a lot of study data are available concerning the diagnostic accuracy of liver fat quantification compared with histology or MRI. Terminology of the technique depends on the manufacturer. Sensitivity for detection of steatosis ranges from 68 to 88% with a specificity from 62 to 100% [27,28,29], depending on the study, method and chosen cut-off value. Similar to B-mode sonography, the diagnostic accuracy for higher grades of steatosis is also better for the new quantitative methods. A study using the ultrasound-derived fat fraction (UDFF) technology from Siemens is highlighting, because the diagnostic accuracy for the detection of steatosis was 0.97 (sensitivity 94%, specificity 100%) [30]. As a limitation, it should be noted that the number of patients included was small and only six patients did not have hepatic steatosis. It remains to be seen whether these results can be extrapolated to a larger population. To date, there have been no comparative studies demonstrating the superiority of one manufacturer’s technology over another.

In summary, the new quantitative ultrasound-based techniques are also excellent in detection of moderate and severe steatosis. However, the diagnostic accuracy for mild steatosis is fair but not excellent, so that the mentioned diagnostic gap cannot be completely closed. Table 1 shows the performance of the mentioned diagnostic tools for detection of steatosis and their advantages and disadvantages.

Why is it important to quantify liver steatosis?

First, it is important to detect abnormal steatosis so that the diagnosis of fatty liver disease can be made. Second, it is known that higher grade of steatosis is a risk factor of fibrosis progression [31] so that high risk patients can potentially be identified. Third, repeated quantitative measurements of fat content could play a role in the follow-up and could represent an additional motivation for the patient to continue lifestyle modifications, although the latter hypothesis has not been proven in systematic studies. However, special caution is required for the follow-up: A reduction in the liver fat content is not necessarily associated with an improvement of the disease. It is known that, in case of progressive disease, liver fibrosis increases, but liver fat content decreases. Therefore, the follow-up should always include a determination of the degree of fibrosis.

## 3. Liver Fibrosis

The degree of fibrosis is the decisive parameter for the prognosis, mortality, and hepatic and extrahepatic events. The risk of severe liver disease increased with higher stage of fibrosis. Patients with NAFLD cirrhosis have predominantly liver-related events, whereas those with bridging fibrosis have predominantly non-hepatic cancers and vascular events [32,33,34]. Histology is still the gold standard for determining the degree of fibrosis. Five stages of liver fibrosis are distinguished according to severity (F0–F4) [35]. However, a liver biopsy is required with the above-mentioned disadvantages.

To determine the grade of liver fibrosis, it makes sense to use non-invasive procedures. The basic imaging examination is sonography of the liver. In advanced chronic liver disease, characteristic changes in the liver such as a nodular liver surface, inhomogeneous parenchyma (Figure 2), a rounded inferior border of the liver, a hypertrophic left liver lobe, and changes in the liver vessels borders can be observed. For the diagnosis of liver cirrhosis, these findings have a high specificity (82–100%), but a lower sensitivity (20–91%) [36]. Conversely, this means that advanced cirrhosis in presence of the above changes can be reliably detected by using B-mode sonography. However, if these criteria are missing, an exclusion of cirrhosis is not possible. To determine significant fibrosis (F ≥ 2), the diagnostic accuracy of sonography is insufficient (53–61%), even when several parameters are combined [37].

### Elastography

Since 2002, a device has existed that makes it possible to measure the stiffness of liver tissue quantitatively. This technique revolutionized sonography in hepatology. The so-called FibroScan from Echosens measures the speed (unit m/s) of shear waves generated by a mechanical impulse which is then transmitted through the liver tissue. Shear waves are slow-moving waves that are perpendicular to the direction of propagation. They run slower through soft tissue, and through hard tissue they run more quickly. The speed of the shear waves is a parameter of liver stiffness, and liver stiffness usually shows a good correlation with the degree of fibrosis of the liver. This method is called transient elastography (TE). TE is an established method for determining the degree of fibrosis in chronic liver diseases [38]. In >1300 publications, this technique has been tested in various liver disease etiologies, and cut-off values for stages of fibrosis were determined using histology as the reference standard. The diagnostic accuracy for fibrosis grade F ≥ 2 is good (88%), for F ≥ 3 (91%) and F = 4 very good (93%) [39]. This classification is best validated in patients with chronic hepatitis B and C. The application of TE is not possible in case of perihepatic ascites, very obese patients, and narrow intercostal spaces. In addition, several studies have identified factors that lead to greater liver stiffness values without evidence of higher degree of fibrosis, such as intrahepatic cholestasis, highly elevated alanine aminotransferase (ALT) values as an indication of inflammation, or a congested liver in case of right heart failure. The FibroScan was developed specifically to carry out the TE. A sonographic examination of the liver is therefore not possible.

In the last years, almost all manufacturers of sonography equipment have started to integrate elastography functions into their sonography devices. These functions were called point shear wave elastography (pSWE) and 2D-SWE (2-dimensional shear wave elastography), respectively. 

In contrast to TE, the shear waves are not caused by a mechanical impulse but by an acoustic radiation force impulse (ARFI). Pressure pulse is generated, and its speed is measured in a small area (point) or in an area of a few centimeters’ length (2D).

This area (also known as the “region of interest” (ROI)) is defined before the measurement by the B-mode examiner and is placed within the liver parenchyma, free of larger vessels or structures. After delivery of the acoustic pressure pulse, the area is color coded (usually blue = low shear wave velocity, red = high shear wave velocity). After 1 to 2 s there is an automatic new measurement [40]. As soon as an image with homogeneous color coding has been generated, the investigator selects an area within the color box in which the shear wave velocity in m/s is displayed. This value can be converted via the young Modulus to a value in the unit kilopascal (kPa). An example of one representative liver stiffness measurement is shown in Figure 3. Several individual measurements, usually 10 to 12 in number, are then performed and the median is calculated. Although both TE and 2D-SWE use the shear wave velocity as a measure of the underlying tissue stiffness, cutoff values from the well-described transient elastography cannot be straightforwardly applied to 2D-shear wave elastography (2D-SWE) and pSWE. This fact represents a major limitation for these newer methods. Importantly, reference values of 2D-SWE differ among manufacturers, because each manufacturer uses their own patented methods for the calculation [40]. 

Regardless of the elastography method, the cut-off values for the fibrosis grades are also dependent on the underlying etiology of the liver disease [38]. While initially only studies with patients with chronic viral hepatitis were available, to date there has been good data on the use of the various elastography methods in patients with NAFLD. A recent meta-analysis identified 54 studies used TE, 11 studies used pSWE, and 4 studies used 2D-SWE for evaluation of liver fibrosis in patients with NAFLD [41]. It should be noted that with regard to the 2D-SWE, only studies in which the Aixplorer device (SuperSonic Imagine, Aix-en-Provence, France) was used were included. Reference standard in all included studies for the diagnosis of NAFLD and for grade of fibrosis was histology. The diagnostic accuracy in detecting significant fibrosis (≥F2) were for TE 0.83 with a sensitivity of 80% and a specificity of 73%; for pSWE 0.86 (sensitivity 69%, specificity 85%); and for 2D-SWE: 0.75 (sensitivity 71%, specificity 67%). 

The diagnostic accuracy in detecting advanced fibrosis (≥F3) were for TE 0.85, with a sensitivity of 80% and a specificity of 77%; for pSWE: 0.89 (sensitivity 80%, specificity 86%); and for 2D-SWE 0.72 (sensitivity 72%, specificity 72%).

The diagnostic accuracy in detecting cirrhosis (F4) were for TE 0.89 with a sensitivity of 76% and a specificity of 88%; for pSWE: 0.90 (sensitivity 76%, specificity 88%); and for 2D-SWE: 0.88 (sensitivity 78%, specificity 84%). 

The cut-off-values of all elastography methods for the stages of fibrosis have a wide range with large overlaps, and validation of pre-specified cut-offs in prospective studies are lacking [41].

Another large meta-analysis which compared laboratory tests, ultrasound, or magnet resonance elastography (MRE) to detect fibrosis in patients with non-alcoholic fatty liver disease revealed that shear wave elastography (no distinction was made between 2D-SWE and pSWE in this meta-analysis) had higher diagnostic accuracy for staging fibrosis than TE [42].

A recent study could show that inflammatory activity on histology significantly affects liver stiffness value by TE, but not by 2D-SWE in patients with NAFLD. As a possible consequence of this observation, also in this study, 2D-SWE reflects liver fibrosis more accurately than TE [43].

A potential advantage of 2D-SWE compared to the TE is that with the use of 2D-SWE, a significantly larger area of the liver is examined, and this area is more representative for the entire organ. 

A particular problem in NAFLD patients is the failure rate of elastography. Analogous to the CAP measurement, different studies report increased failure rates owing to increased body mass index (BMI > 30 kg/m²), intercostal wall thickness >25 mm or waist circumference >102 cm, which may interfere with the transmission of the push impulses and the tracking ultrasound, thus preventing correct estimation of liver stiffness [38,44,45,46]. The additional use of the XL probe resulted in a marked increase in the success rate of valid measurements in real-life cohorts [47]. However, the XL probe is only available for TE. In one study, a success rate of only 63% was determined in a difficult-to-scan cohort using 2D-SWE (Aixplorer, Supersonic imagine, Aix-en-Provence, France), whereas TE (XL-probe) was possible in 96% of the patients [46]. According to the literature, the range of technical failure of the elastography methods in patients with NAFLD was wide. The percentage of invalid measurements reached from 2% to 49% using TE, 0–43% using pSWE and 3–27% using 2D-SWE. In up to 30% of the studies, failure rates were not reported [41]. A possible explanation of these variability could be the fact that quality criteria for a valid measurement using shear wave elastography are absent (depending on the manufacturer) or have not been satisfactorily defined. Thus, a subjective component of the examiner could influence the measurement, because the examiner decides in which elastogram a measurement is made and in which not [40]. Most studies using 2D-SWE worked analogous to following criteria: The elastograms were considered representative and reliable only if they fulfilled the following quality criteria: (1) more than two thirds of the elastographic map had to be homogenously colored or have gradual color transition; (2) artifacts (spots/pixelization/lack of signal) had to occupy less than one third of the elastographic map and (3) no sharp transition from soft (blue) to hard (red) elastographic areas was permitted. Patients without reliable measurements according to these criteria were called ‘non-successful’ because of technical failure. In addition, patients who had measurements with an interquartile range (IQR) above 30% of the median value were also called ‘non-successful’ because of unreliable results [48]. Some other manufacturers have already developed more objective criteria [38]. 

Another practical problem in clinical practice could be the availability of the elastography methods. Sufficient data about availability are lacking; however, it can be assumed that TE can only be offered in specialized centers due to the limitation that no B-mode sonography can be performed with the Fibroscan and therefore an elastography method is used in most gastroenterological and hepatological clinics and practices, which is integrated into a high-end ultrasound device. As shown, for many elastography devices using 2D-SWE in patients with NAFLD, valid data for determining fibrosis with histology as the reference standard are missing. Furthermore, in resource-limited developing countries, the availability of elastography-capable sonography devices is limited [49].

Table 2 shows the performance of diagnostic tools for significant fibrosis (F ≥ 2), advanced fibrosis (F ≥ 3), cirrhosis (F = 4), and their advantages and disadvantages. The blood-based tests FIB-4, NFS, and APRI (Aspartate aminotransferase-to-platelet ratio index), which are not the focus of this review, are also listed here.

In summary, the development of elastography revolutionized sonography in hepatology so that the grade of fibrosis can be determined with a non-invasive technique with good diagnostic accuracy. The main aim is to identify patients with advanced fibrosis or cirrhosis. The main problems for clinical practice seem to be the limited availability of TE, lack of data for 2D-SWE of some ultrasound devices and the high failure rate of liver stiffness measurements in obese patients, especially for 2D-SWE.

## 4. Screening for Hepatocellular Carcinoma

Worldwide hepatocellular carcinoma (HCC) arises in a cirrhotic background in up to 90% of cases. The highest prevalence of HCC is in sub-Saharan Africa and eastern Asia [51]. Surveillance by biannual ultrasound is recommended for such patients because it allows diagnosis at an early stage when effective therapies are feasible. A meta-analysis including 19 studies showed that ultrasound surveillance detected the majority of HCC tumors before they presented clinically, with a pooled sensitivity of 94%. Ultrasound was less effective for detecting early-stage HCC, with a sensitivity of only 63% [52]. The role of surveillance for patients with NAFLD without cirrhosis is unclear [53]. The annual incidence of NAFLD-related HCC is expected to increase by 45–130% by 2030. Diabetes mellitus is the most important risk factor for HCC development in NAFLD. The highest risk of HCC exists in patients with advanced fibrosis or cirrhosis, although 20–50% of HCC cases arise in NAFLD patients with an absence of cirrhosis. Although NAFLD patients without advanced fibrosis have a higher risk of developing HCC than healthy controls, the absolute individual risk is low, so a general screening is not recommended [54,55]. The sensitivity of ultrasound for detecting HCC can be reduced in NAFLD patients: About 20% of ultrasound examinations for patients with cirrhosis are of inadequate quality to rule out the presence of HCC, mostly due to ultrasound artifacts, inadequate ultrasound penetration and patient-related characteristics, such as obesity [56]. The likelihood of inadequate ultrasound quality is significantly higher in overweight or obese patients [57]. Furthermore, tumor-related factors, such as subcapsular location, small size, and infiltrative tumor type, can significantly impair the sensitivity of ultrasound [58]. In this context, another important issue is the quality of the ultrasound equipment and the experience of the examiner. Recent guidelines for HCC surveillance in patients with NAFLD recommend that future screening should be performed by either computed tomography (CT) or a magnetic resonance imaging scan when the quality of ultrasonography is suboptimal for the screening of HCC [57]. 

## 5. Conclusions

Ultrasound is the primary imaging modality in the assessment of patients with confirmed or suspected NAFLD. The fist aim is to identify patients with NAFLD and to make the diagnosis. B-Mode ultrasound is an excellent method to detect patients with moderate and severe steatosis, whereas the sensitivity for mild steatosis is low. HRI and quantitative ultrasound methods based on attenuation parameters can improve the identification of patients with mild steatosis. The second aim is to identify patients with advanced fibrosis and cirrhosis. B-mode ultrasound has high specificity but low sensitivity in this regard. The development of elastography methods was a milestone in non-invasive assessment of liver fibrosis. TE and the newer methods pSWE and 2D-SWE have good diagnostic accuracy to determine the grade of fibrosis. However, especially in obese NAFLD patients, the rate of invalid measurement is quite high. NAFLD patients with advanced fibrosis or cirrhosis have an increased risk for HCC. Ultrasound is the primary imaging modality for HCC screening. However, especially in obese NAFLD patients, this method has potential limitations, and in cases of suboptimal imaging quality, CT or MRI are recommended.

## Figures and Tables

**Figure 1 jcm-11-04581-f001:**
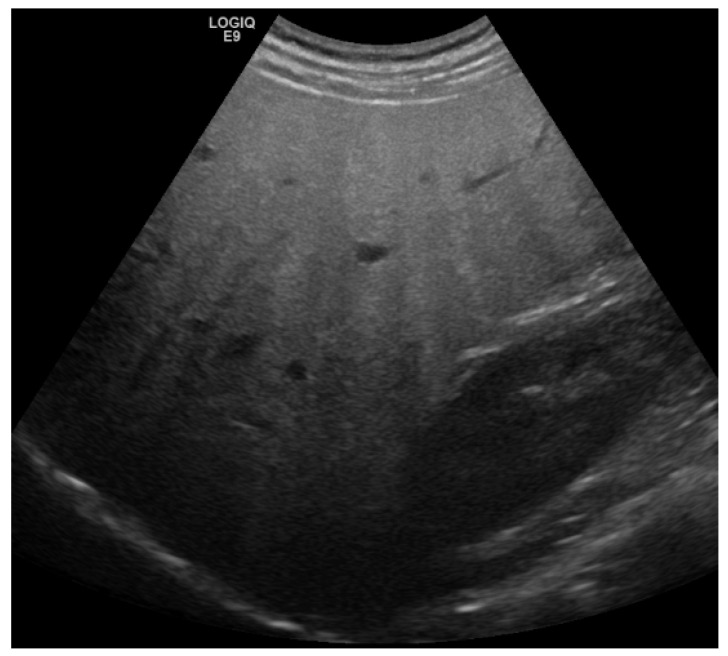
B-Mode sonography with marked increase in fine echoes with poor or non-visualization of the intrahepatic vessel borders, diaphragm, and posterior right lobe of the liver. This finding is pathognomonic for steatosis.

**Figure 2 jcm-11-04581-f002:**
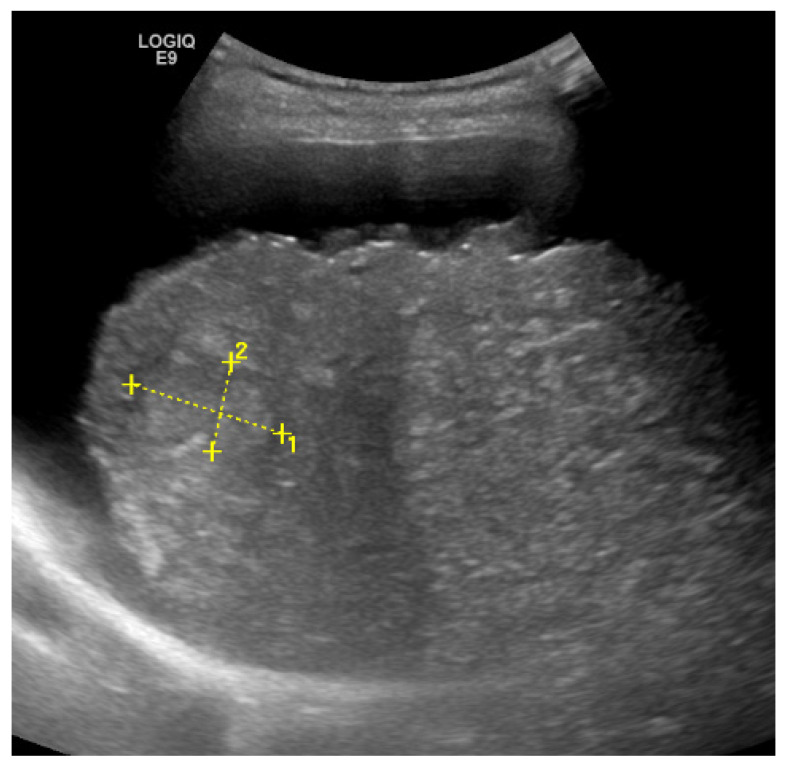
Characteristic changes in the liver. It shows nodular liver surface, perihepatic ascites, and inhomogeneous parenchyma in a NAFLD patient. These findings are pathognomonic for cirrhosis. The lesion in the right lobe of the liver (dimension 1 and 2) is suspicious for HCC.

**Figure 3 jcm-11-04581-f003:**
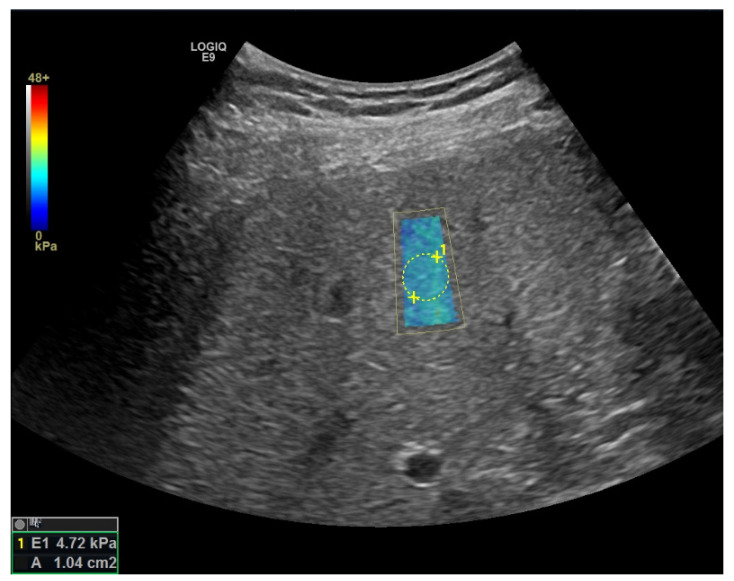
Representative liver stiffness measurement in a NAFLD patient with only simple steatosis. The elastogram fulfilled the quality criteria.

**Table 1 jcm-11-04581-t001:** Shows the performance of diagnostic tools for detection of steatosis and their advantages and disadvantages.

	Sensitivity	Specificity	Advantage	Disadvantage
B-Mode US [14,15,16,17,18]	53–76%	76–93%	-High availability -high-end-device not required	-semiquantitative -low sensitivity for mild steatosis (<30%)
HRI [20,21,22,23]	62.5–100%	54–95%	-quantitative -potentially better than US alone	-low sensitivity for mild steatosis (<30%) -additional program required to calculate HRI -lack of evidence, HRI-value depends on US device and cause of liver disease
CAP [24]	68.8%	82.2%	-widely validated, high evidence -defined cut-off values for grades of steatosis	-dedicated device required -lower accuracy in very obese patients
Newer fat-quantification techniques [27,28,29,30]	(68–100%)	(62–100%)	-integrated in high-end devices -potentially high sensitivity (more studies required)	-low evidence, lack of studies

US: ultrasound; HRI: hepatorenal index; CAP: controlled attenuation parameter.

**Table 2 jcm-11-04581-t002:** Shows the performance of diagnostic tools for significant fibrosis (F ≥ 2), advanced fibrosis (F ≥ 3), and cirrhosis (F = 4), and their advantages and disadvantages.

	Sensitivity	Specificity	Advantage	Disadvantage
B-Mode US [36,37]	F ≥ 2: 32% F ≥ 3: 40% F = 4: 20–91%	F ≥ 2: 85% F ≥ 3: 85% F = 4: 82–100%	-high availability -high specificity for cirrhosis	-high US experience required, -low sensitivity, even for cirrhosis -very low accuracy for fibrosis stages
TE [41]	F ≥ 2: 80% F ≥ 3: 80% F = 4: 76%	F ≥ 2: 73% F ≥ 3: 77% F = 4: 88%	-widely validated, high evidence -application recommended in guidelines -defined quality criteria -XL-probe for obese patients	-dedicated device required (low availability outside of centers)
pSWE [41]	F ≥ 2: 69% F ≥ 3: 80% F = 4: 76%	F ≥ 2: 85% F ≥ 3: 86% F = 4: 88%	-integrated in high-end devices, performing in combination with regular US	-smaller ROI
2D-SWE [41]	F ≥ 2: 71% F ≥ 3: 72% F = 4: 78%	F ≥ 2: 67% F ≥ 3: 72% F = 4: 84%	-integrated in high-end devices, performing in combination with regular US -larger ROI (potential more representative)	-high failure rate in obese patients -lack of studies for most devices -lack of defined quality criteria
FIB-4 [50]	F ≥ 3: 69%	F ≥ 3: 70%	-based on simple variables widely available in clinical practice -free online calculator -Optimized-Cut-off Value: High NPV for ruling out advanced fibrosis	-High risk of false positive results for advanced fibrosis
NFS [50]	F ≥ 3: 75%	F ≥ 3: 63%		-High risk of false positive results for advanced fibrosis -lower performance in obese and diabetic patients
APRI [50]	F ≥ 3: 67%	F ≥ 3: 63%	-only 2 simple parameters required -free online calculator	-Lower performance than FIB-4 and NFS

US: ultrasound; TE: transient elastography; pSWE: point shear wave elastography; 2D-SWE: 2-dimensional shear wave elastography; FIB-4: Fibrosis-4 Index; NFS: NAFLD Fibrosis Score; APRI: AST-to-platelet ratio index; AST: aspartate aminotransferase; ROI: Region of interest; NPV: negative predictive value.

## Data Availability

Not applicable.

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
