# Peer review of "Role of Ultrasound Methods for the Assessment of NAFLD"

_jcm, 2022, doi:10.3390/jcm11154581_

Round 1
Reviewer 1 Report
Journal of Clinical Medicine
Role of ultrasound methods for the assessment of NAFLD
In the manuscript, the authors produced a literature review focusing on different ultrasound methods to detect and graduate hepatic steatosis and to determine the grade of fibrosis in patients with NAFLD. The theme is relevant and sheds some light on NAFLD. However, I have some major and minor concerns.
Major concerns
The authors did not define what is the type of their review: it is narrative or a systematic review?
The main objective of a systematic review is to formulate a well-defined research question and use qualitative and quantitative methods to analyze all the available evidence attempting to answer the question. In contrast, narrative reviews can address one or more questions with a much broader scope.
There is several repeated information in the subsections.
The manuscript should be extensively English proofread.
Minor concerns
Keywords: 2DSWE and NAFLD; what mean?
- It is desirable that the cited keywords are present in the abstract.
Introduction
NASH.
- There are several terms in the entire manuscript without meaning, please, revise the whole manuscript.
A recent nationwide study revealed.
- In what country the study was conducted?
cardiovascular disease and hepatocellular carcinoma (HCC)[5].
- The term (HCC) appeared previously
EASL:
- What is the meaning?
According to the current EASL guideline, magnet resonance imaging (MRI) is now considered equivalent to biopsy to determine the degree of liver fat content
- Please, add a reference for the paragraph
The following article is intended to provide a practice-oriented overview...
- I suggest: Our aim is to provide a practice-oriented overview of the use of sonography methods for the assessment of patients with NAFLD.
2. Detection and graduation of hepatic steatosis
s (i.a. impaired visualization of diaphragm), the specificity of the presence of steatosis is nearly 100%
- i.e.?
To close this gap, new ultrasound methods were recently developed.
- I suggest: solving, elucidating, etc.
BMI, CAP-value
- what is the meaning?
Extensive paragraphs: Clinical studies that have used tools based on the envelope statistics of the backscattered ultrasound are those performed by using the acoustic structure quantification or other parameters derived from it, such as the normalized local variance, and that performed by estimating the speed of sound[24].
- In the manuscript, there are several extended paragraphs that should be shortened.
A current position paper of the World Federation for Ultrasound in Medicine & Biology (WFUMB)
- The reference must be added to the bibliography
Indeed, this thesis has not been proven in systematic studies
- What means this thesis?
Although both TE and 2D-SWE use the shear wave velocity as a measure of the underlying tissue stiffness, cutoff values from the well-described transient elastography cannot be straightforwardly applied to 2D-shear wave elastography (2D-SWE) and pSWE which represents a major limitation for these newer methods.
- Extense paragraph: > 40 WORDS
The cut-off-values of all elastography methods for the stages of fibrosis have a wide range with large overlaps and validation of pre-specified cut-offs in prospective studies is lacking.
- Plese cite bibliograhy
An objectifiable advantage
¬ objectifiable?
An objectifiable advantage of 2DSWE compared to the TE is that with 2DSWE a significantly larger area in the liver is examined and this area is more representative of the entire organ
- Is this technology available in developing countries? What is the cost?
A particular problem in NAFLD patients is the failure rate of elastography. Different studies report increased failure rates owing to increased body mass index (BMI > 30 kg/m²), intercostal wall thickness >25 mm or waist circumference >102cm, which may interfere with the transmission of the push impulses and the tracking ultrasound, thus preventing correct estimation of liver stiffness[36,42–44].
- This information was previously given
4. Screening for Hepatocellular carcinoma
In the Western world.
The Western world is formed by which countries? How about Africa, Asia, and the Americas?
Bibliography
- Must be adjusted and standardized according to the M&M
Journal of Hepatology
Am J Gastroenterol
Author Response
Dear Editor and Reviewer,
Thank you for giving us the opportunity to submit a revised version of the manuscript “Role of ultrasound methods for the assessment of NAFLD” by Petzold for consideration of publication in Journal of Clinical Medicine.
I thank the reviewers for their thorough assessment of the manuscript and the constructive criticism. I have addressed all comments point by point and carefully revised the manuscript accordingly.
I feel that the revised version of the manuscript has gained in strength and clarity and hope it will now be suitable for publication in Journal of Clinical Medicine.
I look forward hearing from you.
Sincerely,
Golo Petzold
Response to Reviewer 1:
Point 1: The authors did not define what is the type of their review: it is narrative or a systematic review?
The main objective of a systematic review is to formulate a well-defined research question and use qualitative and quantitative methods to analyze all the available evidence attempting to answer the question. In contrast, narrative reviews can address one or more questions with a much broader scope.
Response: I thank the reviewer for raising this point. It is a narrative review. In the revised manuscript it is mentioned in the abstract.
Point 2: There is several repeated information in the subsections.
Response: I thank the reviewer for this point. Of course, Abstract, Introduction and Conclusion provide similar information. In the revised version of the manuscript, in the other subsections some text passages have been adjusted.
Point 3: The manuscript should be extensively English proofread.
Response: Linguistic changes were made in the revised manuscript.
Point 4: Keywords: 2DSWE and NAFLD; what mean?
- It is desirable that the cited keywords are present in the abstract.
Response: I thank the reviewer for this important point. In the revised manuscript, the Keywords were explained and mentioned in the abstract.
Point 5: NASH.
- There are several terms in the entire manuscript without meaning, please, revise the whole manuscript.
Response: Thank you for this point. In the revised version of the manuscript, I added the lacking meanings.
Point 6: A recent nationwide study revealed. - In what country the study was conducted?
Response: The study was conducted in Sweden. In the revised version of the manuscript this information is mentioned (line 39).
Point 7: cardiovascular disease and hepatocellular carcinoma (HCC)[5]. - The term (HCC) appeared previously
Response: Thank you. In the revised manuscript this error has been corrected.
Point 8: EASL: - What is the meaning?
Response: In the revised manuscript the meaning was added.
Point 9: According to the current EASL guideline, magnet resonance imaging (MRI) is now considered equivalent to biopsy to determine the degree of liver fat content - Please, add a reference for the paragraph
Response: The lacking reference was added to the revised manuscript.
Point 10: The following article is intended to provide a practice-oriented overview... - I suggest: Our aim is to provide a practice-oriented overview of the use of sonography methods for the assessment of patients with NAFLD.
Response: I thank the reviewer for this good advice. In the revised manuscript the suggestion was accepted.
Point 11: Detection and graduation of hepatic steatosis s (i.a. impaired visualization of diaphragm), the specificity of the presence of steatosis is nearly 100% - i.e.?
Response: I thank the reviewer for this point. To clarify this statement, the sentence has been adjusted as follows: “In presence of sonographic criteria of a higher-grade steatosis (i.a. impaired visualization of diaphragm), the probability of the presence of hepatic steatosis is nearly 100%”
Point 12: To close this gap, new ultrasound methods were recently developed.
- I suggest: solving, elucidating, etc.
Response: I thank the reviewer for this good advice. In the revised manuscript the suggestion was accepted
Point 13: BMI, CAP-value - what is the meaning?
Response: Thank you for this point. In the revised version of the manuscript, I added the lacking meanings.
Point 14: Extensive paragraphs: Clinical studies that have used tools based on the envelope statistics of the backscattered ultrasound are those performed by using the acoustic structure quantification or other parameters derived from it, such as the normalized local variance, and that performed by estimating the speed of sound[24]. - In the manuscript, there are several extended paragraphs that should be shortened.
Response: I thank the reviewer for raising this point. In the revised Manuscript, the extended paragraphs were shortened.
Point 15: A current position paper of the World Federation for Ultrasound in Medicine & Biology (WFUMB) - The reference must be added to the bibliography
Response: In the revised manuscript, the missing reference was added.
Point 16: Indeed, this thesis has not been proven in systematic studies - What means this thesis?
Response: I thank the reviewer for this point. To clarify this statement, in the revised manuscript the sentence has been adjusted as follows: “Third, repeated quantitative measurements of fat content could play a role in the follow-up and could represent an additional motivation for the patient to continue lifestyle modifications. However, the latter hypothesis has not been proven in systematic studies.”
Point 17: Although both TE and 2D-SWE use the shear wave velocity as a measure of the underlying tissue stiffness, cutoff values from the well-described transient elastography cannot be straightforwardly applied to 2D-shear wave elastography (2D-SWE) and pSWE which represents a major limitation for these newer methods. - Extense paragraph: > 40 WORDS
Response: I thank the reviewer for this advice. The paragraph was shortened.
Point 18: The cut-off-values of all elastography methods for the stages of fibrosis have a wide range with large overlaps and validation of pre-specified cut-offs in prospective studies is lacking. - Please cite bibliography
Response: In the revised manuscript, the lacking reference was added.
Point 19: An objectifiable advantage ¬ objectifiable?
Response: I thank the reviewer for raising this point. You are totally right, this is misleading. In the revised manuscript the sentence has been adjusted as follows: “A potential advantage of 2D-SWE compared to the TE is that with the use of 2D-SWE, a significantly larger area in the liver is examined and this area is more representative for the entire organ”.
Point 20: An objectifiable advantage of 2DSWE compared to the TE is that with 2DSWE a significantly larger area in the liver is examined and this area is more representative of the entire organ - Is this technology available in developing countries? What is the cost?
Response: I thank the reviewer for raising this important point. Regarding availability of elastography devices in developing countries there is a lack of data. In the revised manuscript a short paragraph was added (line 324-333).
Point 21: A particular problem in NAFLD patients is the failure rate of elastography. Different studies report increased failure rates owing to increased body mass index (BMI > 30 kg/m²), intercostal wall thickness >25 mm or waist circumference >102cm, which may interfere with the transmission of the push impulses and the tracking ultrasound, thus preventing correct estimation of liver stiffness[36,42–44]. - This information was previously given
Response: I thank the reviewer for this point. In the revised version of the manuscript, the second paragraph was shortened (line 288-294). However, TE and CAP are two different techniques and they have not necessarily the same limitations. The existing analogy of the limitations was clarified.
Point 22: Screening for Hepatocellular carcinoma In the Western world.
The Western world is formed by which countries? How about Africa, Asia, and the Americas?
Response: I thank the reviewer for raising this point. This is misleading. In the revised manuscript the sentence has been adjusted as follows: Worldwide hepatocellular carcinoma (HCC) arises in a cirrhotic background in up to 90% of cases. The highest prevalence of HCC is in sub-Saharan Africa and eastern Asia .
Point 23: Bibliography - Must be adjusted and standardized according to the M&M
Journal of Hepatology
Am J Gastroenterol
Response: I thank the reviewer. In the revised manuscript the reference style has been adjusted using Zotero and choosing the style “Journal of Clinical Medicine”.

Reviewer 2 Report
The manuscript (jcm-1802974) entitled "Role of ultrasound methods for the assessment of NAFLD” summarized well for the current tools such as B-mode US, fibroscan, etc. for assessment of NAFLD in terms of steatosis, and fibrosis. Please revise the paper according to my comments.
1. Please provide a table of diagnostic tools: summary of their advantage, disadvantage, sensitivity, and specificity for steatosis and fibrosis.
2. I recommend that it would be better to describe associations of biomarker (such as FIB-4, APRI) and US in NAFLD.
3. The contents of paragraphs A and B below are similar. Please organize it into one.
A: Page 3, “Furthermore, technical limitations of the method are a BMI >35kg/m² and a skin-to-liver capsule distance > 25 mm. These conditions can lead to an overestimation of the CAP values. To solve this problem, later an XL-probe for obese patients was developed. A large metaanalysis showed a similar diagnostic accuracy of XL-probe for detection of steatosis in obese patients compared with the results of M-probe in non-obese patients, whereas the diagnostic accuracy of XL probe seems to be lower in differentiating between no or mild steatosis and higher grades of steatosis[23].
B: Page 7, A particular problem in NAFLD patients is the failure rate of elastography. Different studies report increased failure rates owing to increased body mass index (BMI > 30kg/m²), intercostal wall thickness >25 mm or waist circumference >102cm, which may interfere with the transmission of the push impulses and the tracking ultrasound, thus preventing correct estimation of liver stiffness[36,42–44]. Therefore, the producer of the FibroScan device (EchoSens, Paris, France) developed the XL probe designed for the application in obese patients. The additional use of the XL probe resulted in a marked increase in the success rate of valid measurements in real-life cohorts [45]. However, the XL probe is only available for TE.
Author Response
Dear Editor and Reviewer,
Thank you for giving us the opportunity to submit a revised version of the manuscript “Role of ultrasound methods for the assessment of NAFLD” by Petzold for consideration of publication in Journal of Clinical Medicine.
I thank the reviewers for their thorough assessment of the manuscript and the constructive criticism. I have addressed all comments point by point and carefully revised the manuscript accordingly.
I feel that the revised version of the manuscript has gained in strength and clarity and hope it will now be suitable for publication in Journal of Clinical Medicine.
I look forward hearing from you.
Sincerely,
Golo Petzold
Response to Reviewer 2:
Point 1: Please provide a table of diagnostic tools: summary of their advantage, disadvantage, sensitivity, and specificity for steatosis and fibrosis.
Response: I thank the reviewer for this very good advice. In the revised version of the manuscript, the two suggested tables are added.
Point 2: I recommend that it would be better to describe associations of biomarker (such as FIB-4, APRI) and US in NAFLD.
Response: I thank the reviewer for this important point. Although blood-based tests for staging fibrosis are not in the focus of this narrative review, in the revised manuscript they are mentioned and a comparison with ultrasound techniques was made (line 70-75 and Table 2).
Point 3: The contents of paragraphs A and B below are similar. Please organize it into one.
A: Page 3, “Furthermore, technical limitations of the method are a BMI >35kg/m² and a skin-to-liver capsule distance > 25 mm. These conditions can lead to an overestimation of the CAP values. To solve this problem, later an XL-probe for obese patients was developed. A large metaanalysis showed a similar diagnostic accuracy of XL-probe for detection of steatosis in obese patients compared with the results of M-probe in non-obese patients, whereas the diagnostic accuracy of XL probe seems to be lower in differentiating between no or mild steatosis and higher grades of steatosis[23].
B: Page 7, A particular problem in NAFLD patients is the failure rate of elastography. Different studies report increased failure rates owing to increased body mass index (BMI > 30kg/m²), intercostal wall thickness >25 mm or waist circumference >102cm, which may interfere with the transmission of the push impulses and the tracking ultrasound, thus preventing correct estimation of liver stiffness[36,42–44]. Therefore, the producer of the FibroScan device (EchoSens, Paris, France) developed the XL probe designed for the application in obese patients. The additional use of the XL probe resulted in a marked increase in the success rate of valid measurements in real-life cohorts [45]. However, the XL probe is only available for TE.
Response: I thank the reviewer for this point. In the revised version of the manuscript, the second paragraph was shortened (line 294-298). However, TE and CAP are two different techniques and they have not necessarily the same limitations. The existing analogy of the limitations was clarified.

Round 2
Reviewer 1 Report
Thanks for the opportunity to act as a reviewer for the Journal of Clinical Medicine. Kind regards.